# Eldercare’s Turnover Intention and Human Resource Approach: A Systematic Review

**DOI:** 10.3390/ijerph20053932

**Published:** 2023-02-22

**Authors:** Rahimah Jurij, Ida Rosnita Ismail, Khadijah Alavi, Rokiah Alavi

**Affiliations:** 1Graduate School of Business, Universiti Kebangsaan Malaysia, Bangi 43600, Malaysia; 2Centre for Research in Psychology and Human Well-Being, Faculty of Social Sciences and Humanities, Universiti Kebangsaan Malaysia, Bangi 43600, Malaysia; 3Kuliyyah of Economics and Management Science, International Islamic University Malaysia, Kuala Lumpur 50728, Malaysia

**Keywords:** turnover intention, eldercare, social enterprise, HR, systematic review

## Abstract

Eldercare workers’ turnover intentions have caused serious concern given their high demand and pivotal role in ensuring elderly individuals’ well-being. This systematic review examined the main factors of eldercare employees’ turnover intentions with the purpose of identifying gaps and structure a novel human resource (HR) approach framework for eldercare social enterprises through a global literature review and realistic conclusions. A total of 29 publications appeared between 2015 and 2021 were digitally extracted from six databases and are extensively discussed in this review. Resultantly, eldercare workers’ turnover intentions were positively impacted by job burnout, low job motivation, and restricted job autonomy. The findings of this study correspond to those of past literature, which highlighted the necessity of thoroughly examining eldercare worker retention practices from an organisational (HR) perspective. Furthermore, the current study outlines the factors influencing eldercare workers’ turnover intentions as well as determine proper HR approaches to mitigate employee turnover issues among eldercare workers for organisational sustainability.

## 1. Introduction

Social enterprises, which globally address social and environmental issues entailing human development enhancement, unemployment, life quality improvement, and social inequality, have gained much popularity following their positive social implications [1,2,3]. According to the literature, a social enterprise denotes shareholders’ sustainable ventures with an emphasis on communal goals rather than profit distribution or maximisation [4,5,6].

Eldercare social enterprises have grown significantly as the population has aged [7,8,9]. In line with [10], the worldwide population is currently witnessing (1) unforeseen shifts in the age structure and (2) a high number of older adults, owing to long lifespans and low fertility levels. On a global scale, the number of elderly individuals aged 65 and above (727 million) could increase two-fold from 9.3% in 2020 to 16.0% (1.5 billion) in 2050. Overall, the rapid population growth between 2020 and 2050 could increase the need for eldercare social enterprises to fulfil older adults’ necessities.

Social enterprise management, including eldercare workers’ service retention, remains challenging, although the rising number of eldercare social enterprises has resulted in job creation and multiple work opportunities [11,12,13,14]. Based on global research, such employees’ turnover rates (an average of 25% to 75%) could adversely impact the daily organizational operations and finances [12,13,14,15,16,17], given the high costs incurred in selecting and training new recruits.

Despite the increasing number of markets needs and job prospects in eldercare social enterprises, global organizations that provide care for the elderly have significant rates of employee turnover [12,13,14]. Following [16] and [18], the eldercare workers’ annual turnover rate in long-term care services has raised much concern. Palpably, this sector reflects high employee turnover rates, low-quality services, and poor organizational performance.

In line with [18,19,20,21], eldercare workers intend to resign due to low manager support, heavy workloads, and low compensation. Considering the important roles of eldercare workers in taking care of elderly patients, it is deemed pivotal to highlight alternatives to enhance their wellbeing and increase their job satisfaction. Malaysia will be an ageing country by 2030 [8,10,22], with elderly care institutions still unable to meet 30.8% of the older population’s daily social support demands [8]. This urgent situation should be addressed strategically in order to ensure the quality of the services provided to older patients while also improving the job performance of eldercare workers.

Therefore, this research aimed to examine the factors influencing eldercare workers’ turnover intentions for optimal organisational retention and eldercare social enterprise expansion. This research highlights the knowledge gap and possible retention approaches for effective HR practices. Notably, 29 publications were reviewed to determine the factors influencing eldercare workers’ turnover intention, actual employee turnover implications at work, and the significance of fulfilling the elderly care sector’s demands. A proposed study framework for the eldercare social enterprise was also established to mitigate the current organisational barriers and optimise organisational retention and expansion.

## 2. Methodology

This section clarifies three main sub-sections: the need for a systematic literature review, the review protocol (ROSES), and the systematic review process. The systematic review process covers identification, screening, eligibility, data extraction, and analysis. This section also contains a list of databases used in this study, as well as search strings and the inclusion and exclusion criteria.

### 2.1. The Need of a Systematic Literature Review

A systematic literature review (SLR) is carried out using transparent and intensive replicable procedures to guide research objectives, reduce research bias, and motivate researchers to find and present high-quality evidence with exceptional findings. Some works [23,24,25] mentioned that a systematic review is a quantitative and qualitative process that involves combining, evaluating, and recognizing relevant data in order to address a research question. Moreover, the systematic literature review methodology initially started in the medical field, it appeared to be useful, and was extended across a variety of study areas, which include engineering, social welfare, education, and law [23].

There is a vast amount of eldercare worker or geriatric workforce turnover literature conducted worldwide. Despite this, a limited, comprehensive, and systematic literature review has been conducted. Due to the fact that addressing retention issues in the geriatric workforce is critical to meeting the current demands of the world’s ageing population, a systematic literature review is required to understand the factors and retention strategies proposed so far in the literature. Hence, to obtain significant results on the main factors of turnover intention among eldercare social enterprise workers, conducting a systematic review is fundamental for assisting future researchers and practitioners in shedding light on the best strategic approaches to managing the human resource involved in the service. In the establishment of this systematic review, the researchers were guided by the key research question, “What are the factors of turnover intention among eldercare workers of eldercare social enterprises?”; most importantly, the main focus of this investigation was to discover the strategic HR approaches to retain eldercare workers in their current workforce.

### 2.2. ROSES as a Review Protocol

ROSES (Reporting Standards for Systematic Evidence Syntheses) was the protocol used for this systematic review. ROSES was established by [23] and attempts to improve the methods for producing systematic literature reviews by maximizing transparency and assuring a quality control. The ROSES-guided SLR procedure will begin with the formulation of research questions using the PICo method: “P” for issue or population, “I” for interest, and “Co” for context. In this study, “P” represents the number of eldercare workers, “I” represents the turnover intention, and “Co” represents the eldercare social enterprise.

Hence, the ROSES review protocol and its review process suit this current review because they can accommodate variation and the diversity of the research subjects, which reflect the heterogeneity or interdisciplinarity of topics and methods [23]. Furthermore, the key strength of ROSES for this study is that it emphasizes the beginning and mid-stages of the review procedure, i.e., searching, screening, data extraction, and quality assessment (see Figure 1). This flexibility is essential and critical for being pertinent to an extensive range of synthesis methods [23,24].

Considering that both Scopus and Web of Science are robust and comprise over 256 research disciplines, inclusive of business and social science studies, the current study’s review procedures were undertaken utilizing these two major databases. In particular, Scopus indexes 763 journals, and Web of Science (Social Science Citation Indexed) indexes 108 journals related to organisational behaviour and human resource management, geriatrics and gerontology, social sciences, and care planning. A previous study [26] proposed that researchers should perform their searches using a high number of databases for the purpose of boosting their chances of identifying relevant publications. Therefore, the current study performed manual searches on other reputable databases, including Science Direct, Microsoft Academic, Springer, Google Scholar, and Semantic Scholar, as these are reputable databases that include business management-related articles.

### 2.3. Systematic Review Process

#### 2.3.1. Identification

Relevant articles were retrieved digitally from six databases using the following keywords: “turnover intentions,” “retention,” “intention to stay,” “aged care workers,” “geriatric,” “eldercare,” “nursing homes,” “social workers,” “ageing care workforce,” “retention rates among eldercare workers,” “nursing homes,” “long-term care,” and “turnover retention rates among eldercare workers.” The first step was the identification of keywords, followed by the process of searching for associated and related terms with the use of a thesaurus, dictionaries, encyclopaedias, and prior studies. Subsequent to identifying the pertinent terminology, search strings for the Scopus and Web of Science databases were formed (refer to Table 1). The first step of the systematic review procedure found 412 articles. In addition, a search process relied on a manual search approach, such as “handpicking” and the snowballing method, was used for other related databases.

#### 2.3.2. Screening

In seven reputable databases, a total of 412 papers were identified for this analysis. After eliminating duplicates, a total of 156 articles were discovered. After including supplementary exclusion criteria, such as non-full-text papers, publication year, and study samples, the articles were assessed for admissibility. Next, 200 articles were rejected in the eligibility stage, since their content did not properly convey the causes of staff turnover or the goal to retain social enterprise employees. At the conclusion of this research, 29 full-text papers were examined.

This review included (i) full-text papers or articles containing empirical data; (ii) articles published between 2015 and 2021; (iii) articles written in English; (iv) articles with samples related to eldercare or the long-term care workforce; and (v) articles with a global country coverage. In contrast, the exclusion criteria for this review were as follows: (i) review articles, conference proceedings, book chapters, non-full-text articles, and duplicate publications; (ii) articles published in 2014 or earlier; (iii) non-English articles; and (iv) articles that did not reflect either the intention to leave or the intention to remain of the eldercare social enterprise workforce (refer to Table 2).

#### 2.3.3. Eligibility

Only 56 (46%) articles proved valid post-screening in stage two, given their association with eldercare employees’ turnover intention factors in a formal work context. It was important at this step for the titles, abstracts, and primary contents of all the articles to be scrutinised to ensure that the inclusion criteria were met and that they were suitable for the current study’s aim of meeting the research objectives. Due to their lack of empirical data and lack of emphasis on the long-term care workforce or geriatric healthcare environment, a total of 27 publications were rejected.

As such, 29 publications were selected as pivotal references to establish the recommended retention strategies for optimal HR practices and empirically support the current research. These papers, which primarily regard nations that currently face or predict a rapid rise of their ageing populations, such as Germany, China, Sweden, Australia, Korea, Malaysia, and Japan [13,14,15,16,17,18,19,20,21,27,28,29,30,31,32,33,34,35,36,37,38,39,40,41,42,43,44,45,46,47,48], were duly reviewed.

Subsequently, a quality evaluation procedure was led using modified criteria suggested in a previous study [48]. This current review predetermined each selected article before incorporating it into the review. The procedure included data extraction and analysis that were led by the research topic, with the collected data analysed using thematic analysis. The authors ensured the suitability of each article in order to align with the aim of the systematic review.

#### 2.3.4. Data Abstraction and Analysis

This study conducted an integrative review, which is one of the review methods that analyse and synthesise various research designs (qualitative, quantitative, and mixed methods) together, by transforming one type into the other—qualifying quantitative data or quantifying qualitative data [49]. This study chose to evaluate all collected data qualitatively. Based on the results of the thematic analysis, relevant themes and subthemes were developed. The first phase of theme creation was data collection.

In this phase, the authors assessed the 29 selected publications for statements or data that provided answers to the study questions. In the second step, the coding approach was employed to organize the data into logical groupings [50]. In the end, the approach revealed three major themes: job burnout, low job motivation, and restricted job autonomy. Subsequently, the authors restarted the procedure for each of the created themes by developing as subthemes any themes, thoughts, or ideas that had a link to the established theme. This additional procedure produced six sub-themes. Furthermore, the authors examined the human resource approaches or solutions highlighted in the selected articles, obtaining four major themes: abilities and competencies, job motivation, improving working cultures, and workplace spirituality, with four sub-themes.

To corroborate the validity of the themes and subthemes, three experts conducted expert evaluations: two were business management and community development experts, while the third was a qualitative expert. The process of expert evaluation established the domain’s validity and ensured the clarity, applicability, and relevance of every sub-theme within its respective themes. Modifications were made in response to the feedback and comments of the experts.

Furthermore, the current study began with an examination of the implications of a dynamic eldercare sector for organisations seeking to retain competent staff and improve service delivery. A brief overview of demographic ageing and its effects on worker–manager jobs’ performances, service quality for older patients, occupational health and safety, and the turnover rates of eldercare workers are also provided. The factors identified in the 29 articles affecting eldercare workers’ turnover intentions, including job burnout, absence of professional skillsets, demotivation, and unconducive work cultures, are extensively discussed.

## 3. Findings

### 3.1. Research Theories or Frameworks

Within the reviewed papers (see Figure 2), 11 (38%) did not involve theories. Frameworks and theories were published in either single or integrated forms. Notably, 11% of the articles employed the job demand–resources model (JD-R), while 7% utilized the conservation of resources theory (CRT) and the two-factor motivation–hygiene theory (TFMH). Due to the obvious lack of theories and frameworks, independent variables were chosen without a theoretical or framework support.

Most of the reviewed studies, which originated from the healthcare or medical science domain, heavily depended on practical and factual research rather than on a theoretical or framework support [47]. Conclusively, JD-R, CRT, and TFMH demonstrated significant applications in turnover intention research, particularly in the context of eldercare. Justifications on the nature of employees or organizations need to be addressed and integrated, respectively, for a high organizational performance.

### 3.2. Region of Study

Most of the eldercare worker’s turnover intention studies were based in Australia, with a total of ten papers [16,19,20,31,33,35,40,42,45,46]. Four papers focused on Korea [15,17,36,47]; three papers covered China [13,38,41]; two papers looked at Japan [21,29]; and two papers focused on Sweden [18,37]. In addition, one paper focused on each of the following countries: Taiwan [48], Hong Kong [30], New Zealand [28], Malaysia [12], Germany [27], Belgium [39], Switzerland [32], Spain [37], and Denmark [34]. 

As for the research methodology, 22 articles from the 29 reviewed articles used a quantitative approach [13,18,19,27,28,29,30,31,32,33,36,37,38,39,40,41,45,46,47,48], while the remaining 5 articles used a qualitative approach [15,16,17,20,35]. Only two studies [34,42] used a mixed-method approach.

In terms of publication year, two articles were published in 2015 and 2016 [35,40], two articles were published in 2016 [35,40], five articles were published in 2017 [12,33,36,42,45], five articles were publish in 2018 [28,29,34,39,46], nine articles were published in 2020 [13,16,17,20,21,30,37,38,47], and two articles were published in 2021 [15,48].

## 4. Discussions

Defined as the preparedness to quit work or the present workplace, “turnover intention” [51,52,53] empirically forecasts the actual staff turnover or resignation [53]. Turnover intentions could function as indicators of actual turnovers following past research [13,54]. Some studies [12,15,55] affirmed the eldercare workers’ high employee turnover rate to be a primary barrier despite its common occurrence. The implications of high employee turnover substantially influenced enterprises’ organisational finances and hampered organisational success when high-performing employees resigned [15,54]. Thus, company HR practices must seriously regard the aforementioned turnover intention factors to minimise employee turnover and the loss of competent workers and improve staff retention.

According to [22], it is critical to address HR management and meet the complexities underlying Malaysia’s ageing population within the healthcare sector. Most scholars conceded the importance of demonstrating a high mental and physical competence to manage elderly’s requirements [15,29,33,42] in terms of emotional stability and physical mobility [29]. Summarily, older adults require eldercare workers’ facilitation to independently perform their daily tasks. For an effective HR approach implementation, eldercare social enterprises must internalise and consider the work nature of eldercare employees.

Based on this study, three main factors influencing eldercare workers’ turnover intentions were determined after reviewing 29 articles. These factors or themes were identified through recurrent themes in selected previous articles and will be summarised under thematic headings [56]; they are job burnout, low job motivation, and restricted job autonomy. Furthermore, researchers identified key themes that emerged from the review process for the suggested HR approaches that could potentially increase the retention rates in eldercare social enterprises, including abilities and competencies, job motivation, improved working cultures, and workplace spirituality (see Table 3 and Table 4).

### 4.1. Job Burnout

Job burnout at work, the first factor, encompasses physical and emotional fatigue [41,57,58]. Eldercare workers with a restless schedule, minimal work–life balance, a high workload, and ongoing shift rotations inevitably suffer from the overwhelming workload and pressure [20,59,60]. Such employees are required to transport elderly patients with little or no movement to the bathroom, to change their diapers, or to conduct physical therapy sessions.

#### 4.1.1. Physical and Emotional Burnout

Physical work commitments could occasionally result in occupational pain or medical conditions involving musculoskeletal disorders [61,62]. The aforementioned concerns induce long-term medical leave and organisational friction following abrupt absences and turnover intentions among eldercare workers. Emotional weariness is caused by a negative work influence, work-life conflicts, low self-confidence as a consequence of inferior competency, verbal or non-verbal abuse from elderly patients, an inadequate work culture, and job discontent [57,63]. The job burnout element constituting physical and emotional exhaustion leads to negative implications that could affect other colleagues, induce an undesirable working environment, and incite workers to resign [62,64]. Notably, the management could encounter challenges in controlling the situation and sustaining organisational operations.

#### 4.1.2. Incompetencies

Incompetent eldercare workers could also experience job burnout. As a capacity for task performance that requires specified knowledge or skills [65,66], competence denotes workers’ ability to fulfil present and future job demands. A study [31] disclosed that high-performing eldercare employees demonstrate low turnover intentions compared to their less competent counterparts. Moreover, low competence could promote low self-confidence and high workplace stress, which make it challenging to manage and perform the assigned tasks [63].

Previous works [55,67] outlined that eldercare services in Asian nations, such as Malaysia, lacked competent workers, which influenced the organisational well-being. The majority of eldercare workers lack the necessary training and educational background to fulfil the requirements of being eldercare workers. As a result, they are prone to making mistakes, are stressed, and struggle to manage their daily tasks appropriately. In summary, low staff competence instigates high turnover rates and undermines organisational service quality.

### 4.2. Low Job Motivation

Low job motivation, the second factor, denotes extrinsic or intrinsic motivation, which is lacking in the eldercare workforce [15,68]. Job motivation is fundamental to increasing employees’ productivity to achieve the organisational goals [13,52]. Employees who experience a high job motivation are likely to feel job satisfaction and perform better in accomplishing the tasks given to them [14,68].

#### 4.2.1. Extrinsic Motivation

According to a study [68], incentives, professional advancement, work security, and performance appraisal characterize the extrinsic employee motivation. Inadequate compensation, work insecurity, and limited career advancement opportunities will influence not only employee’s extrinsic motivation but also their intrinsic motivation [13,69,70]. Meanwhile, [13,15,41] argued that elderly care organisations do not provide their employees with performance appraisals and a clear career path, which leads to job insecurity. In addition, [12,17] contended that eldercare nursing homes offer minimal wages to their employees.

Eldercare employees in countries such as Australia, Korea, Japan, and Malaysia were found to be notably underpaid compared to employees in other sectors, despite their heavy workload and additional tasks. Parallel to what observed in [14,60], the eldercare service struggles to sustain and recruit competent workers following unappealing monetary incentives and poor career progress. Eldercare organisations must comprehensively evaluate the operational expenditures and allocate more resources for employee compensation, benefits, recruitment, and retention, given that almost 40% of eldercare employees considered quitting work due to low salaries, work stress, limited career advancement [15,71], and multi-sector competitors [54,72,73].

#### 4.2.2. Intrinsic Motivation

Regarding intrinsic motivation, [69,70] asserted that workers are intrinsically motivated by professional aspects, including a conducive work setting and relevant job scope, to experience job satisfaction. Although workers could be intrinsically motivated through extrinsic rewards, intrinsic rewards significantly motivate employee behaviour compared to the extrinsic ones. Moreover, individuals’ intrinsic motivation is associated with high work commitment and organizational retention [74]. Low competence and insufficient professional knowledge potentially impact individuals’ intrinsic motivation levels [41,68,74]. Given that most employees suffer from work dissatisfaction and inadequate extrinsic rewards, a low intrinsic motivation is inextricably linked to employee turnover. It is vital to emphasize the improvement of eldercare workers’ intrinsic motivation in organisational HR management, given the paucity of relevant studies.

### 4.3. Job Autonomy

Restricted job autonomy, the third and final factor, could promote a positive working culture. Job or work autonomy, which denotes substantial freedom, independence, and discretion [75], is pivotal for staff welfare, minimal stress, and high employee retention [76,77]. According to other viewpoints, work autonomy could be acknowledged in the form of work schedules, job implementation techniques, working hours, workload, professional goals, and organisational priorities [78,79]. In Ref. [80], the organisational culture constitutes professional values, beliefs, and traditions that potentially improve staff motivation. The organisational culture involves worker–manager interactions, staff empowerment, and decision-making opportunities involving competent co-workers and optimal teamwork [68,80].

#### 4.3.1. Lack of Managerial Support

Following past research, organisational culture-oriented issues, specifically involving staff interactions, low managerial support, social work conflicts, and teamwork, impact employee motivation and escalate turnover intentions [15,16,18,21,40]. Managerial support is essential for fostering a positive work culture and addressing workplace issues, particularly when dealing with the elderly. The authors of [47] and [18] argued that a lack of recognition, not giving employees a chance to voice their opinions, and sharing on-the-job difficulties with managers can lead to trust issues and increase the intention to leave the job.

#### 4.3.2. Teammates Support

Some studies [18,21,32,36] mentioned that eldercare workers experience limited opportunities to discuss work-related challenges with their co-workers. Due to poor management, this issue leads to a lack of support from colleagues in order to complete the daily tasks assigned. They have a better understanding of the organisational goals because the management needs to empower the organisational culture. The top management has to create a better culture in the organisation, especially by addressing social conflicts among staff and managers. When working in the high-demand job of running elderly care organisations, it is crucial to have manager support, help from colleagues, and opportunities to voice out any concerns related to the job routine. Thus, organisations must strive to create a conducive organisational culture that improves job autonomy in eldercare social enterprises.

### 4.4. Human Resource Approach

The current study justifies the main factors influencing eldercare workers’ turnover intentions based on relevant research. Observably, optimal retention strategies are vital to alleviate turnover-oriented concerns. Research on the implementation of organisational HR management approaches remains lacking, despite various suggestions to resolve turnover intentions and sustain eldercare staff.

Based on the reviewed literature, the study framework depicted in Figure 3 led to the conclusion and categorisation of four primary HR approaches (main themes): (1) elevating eldercare workers’ abilities and competencies; (2) reassessing job motivations; (3) improving working cultures; and (4) fostering workplace spirituality. Such alternatives could be holistically applied to local eldercare services and eldercare social enterprises.

#### 4.4.1. Abilities and Competencies

Elderly care social enterprises must first improve their managerial skills to enhance eldercare workers’ competence. Ineffective management instigated turnover intention in the form of inappropriate work shifts, heavy workloads, and low organisational support [14,29,31,32,41,42]. Such complexities could result in physical and emotional exhaustion. Consequently, the HR approaches in eldercare social enterprises could be effectively planned and revised through a top management commitment and organisational efforts [14,68]. In addition to increasing HR management, the management must monitor employees’ competencies and skill sets to assure the quality of eldercare social enterprises and employees’ job performance [14,29,45]. A high work performance demonstrates job satisfaction, which subsequently elevates employee retention.

Training and development effectively ensure eldercare workers’ qualifications and competence to facilitate elderly patients’ daily task performance. Such high-performing employees would be motivated to execute their work assignments. In line with [31], some eldercare employees suffer from job burnout given a lack of adequate skill sets and professional experience [67], which are similar to the limited competency of most eldercare staff in Asian nations. It is deemed pivotal for elderly care social enterprises to develop their skill sets and the capacity to manage heavy workloads and to achieve an emotional balance in handling elderly patients. Hence, qualified eldercare workers who periodically undergo mandatory training could alleviate job burnout and develop their careers.

#### 4.4.2. Reassessing Job Motivation

In reassessing eldercare workers’ extrinsic and intrinsic motivation, the management must provide employees with sufficient remuneration that justly reflects their workload and skills [12,14,16,46]. Alternative benefits, bonuses, yearly increments, security factors, and work promotions should also be regarded in a cost-effective and timely manner [14,27,35,60]. Additionally, the management should meticulously plan employees’ work leaves and workloads to optimally manage their work–life balance and personal conflicts [19,20,27]. Workers would remain motivated with the awareness that their organisation offers personal time off and flexible work schedules. Overall, eldercare workers’ well-being potentially retains their intrinsic motivation and work commitment.

#### 4.4.3. Improving Working Cultures

Supervisors who manage all operational tasks, engage frequently with employees, and serve as mediators for employee-related grievances that need arbitration with the upper management may play a vital role in fostering a healthy workplace and organisational culture in eldercare services [18,21,39,40]. These individuals could positively impact employees’ intrinsic motivation with a conducive environment and establish a social rapport among eldercare workers [29,39,40,41]. The eldercare service supervisors could increase employees’ retention rates and provide professional support for a high task performance [17,18,20,38,40]. Workers are intrinsically motivated and dedicated at work when given the opportunity to be professionally assertive, independent, empowered, and acknowledged [17,18,44,81]. As such, the eldercare supervisors must hone their leadership skills to improve their employees’ retention rates.

#### 4.4.4. Workplace Spirituality

The elderly care social enterprises must strategize to nurture workplace spirituality among eldercare workers. In considering the emotional skills required to manage elderly patients, every eldercare employee must strive to seek values and meaning through their work. 

Similarly, [16,18,21] conceded that meaningful job creation intrinsically stimulates employees to work, given their attachment to the older adults or patients under their care. Such workers may demonstrate high work commitment owing to the intrinsic motivation to serve their organisation and perform value-added tasks that complement their personal goals, apart from high wages or compensation.

The top management should educate their employees on the importance of filial piety, specifically eldercare social enterprises in Asian countries, and the inextricable link to workplace spirituality [82,83,84]. Specifically, people who belong to cultures that respect, honour, and adhere to parents or senior family members prioritise filial piety. The authors of Refs. [36,81,82,85] disclosed that employees who comprehend or fulfil their filial obligations show happiness, minimal work stress, and high intrinsic motivation levels. Such workers, who feel accountable to perform well at work and complete the assigned tasks, would demonstrate low turnover intentions and high retention levels.

Employee turnover intentions could be resolved through workplace spirituality [82,83,86], a passion to serve the elderly, and work resilience [82] for a high organisational retention, according to scholarly perspectives. Workplace spirituality, which may vary from religiosity [87,88], could exist with or without religiosity to develop a positive workplace by establishing meaning and connectedness, supporting workers’ assertiveness, and alleviating inequalities [82,87,89].

The notion of “workplace spirituality”, which promotes employee welfare through high self-assertiveness and workplace morale and ethics [87,88], minimal burnout [82], and optimal co-worker interactions [89,90], has garnered much scholarly attention in business domains. As such, the elderly care social enterprises could nurture workplace spirituality for enhanced productivity and company performance. Eldercare workers should be motivated through workplace spirituality, which would parallel their personal goals and requirements beyond monetary incentives such as attractive wages and benefits [88,89]. Organisations would subsequently optimise their productivity and HR capacity to retain competent staff.

## 5. Conclusions and Recommendations

This study focused on eldercare worker’s turnover intentions and human resources (HR) approaches. It emerged that job burnout, low job motivation, and a lack of job autonomy positively influence eldercare workers’ turnover intentions. This systematic literature review highlighted four appropriate HR approaches for eldercare social enterprises towards organisational sustainability and market relevance: (1) increase eldercare workers’ competence; (2) reassess their extrinsic and intrinsic job motivation through high job security, a career advancement program, and revised compensations; (3) optimise organisational cultures with job autonomy; and (4) nurture and instil workplace spirituality in the organisations. The aforementioned strategies could provide useful insights for the effective implementation of HR approaches in order to accommodate the high eldercare service demands following the rapid growth of the ageing population. Respectively, the HR approach and strategies could lead to an organisational culture change as well as empower the human resource management in eldercare services, which will eventually enhance the service quality. The eldercare services will not only improve workplace culture and employee job performance, but also increase older people’s satisfaction as clients.

Regardless, research on the psychological factors affecting individual welfare and wellbeing in the geriatric workforce or among eldercare workers remains insufficient. This deficiency suggests the need of expanding the present body of research with a focus on enhancing the intrinsic motivation of eldercare workers to supplement the characteristics of eldercare-related jobs, optimistically impact the labour force, and increase employee retention rates. The management of eldercare services could incorporate the aforementioned HR approaches to retain their employees. Nonetheless, such empirical studies primarily highlighted one-sided employee perceptions. It is deemed pivotal to gain a managerial viewpoint on this issue rather than solely relying on eldercare workers’ statements, as HR retention practices are executed by the management. A holistic understanding of managers’ complexities through their professional experience could potentially resolve this workforce conundrum.

Based on the systematic review process of the present study, recommendations for future studies are formulated. First, future scholars could focus on the use of technologies such as artificial intelligence. In accordance with [91], artificial intelligence has empowered eldercare workers to perform nursing tasks efficiently. Most important, empowerment through the artificial intelligence technology will increase employee retention, which is very helpful for social enterprises to strengthen their human resource management. In addition, the artificial intelligence technology could also improve the service quality of social enterprises and reduce the workload burden on the employees [92,93,94]. This is because artificial intelligence machines or robots not only help to support the workload of eldercare workers, but also improve the daily operation of elderly social enterprises. In line with [91], the progress in technology adoption by elderly social enterprises could improve the organisational infrastructure, recruitment, and job advocacy in the elderly care sector, which is in line with the first and ninth goals Sustainable Development Goals (SDGs), i.e., (1) no poverty and (2) industry, innovation, and infrastructure. When job advocacy improves, employability increases, thus, alleviating poverty among potential workers in the ageing sector. Moreover, the public authorities may also implement the outcomes of the study to reinforce policies for aged-care facilities to ensure organisational sustainability.

Second, future scholars could look into green psychology in human resource management in order to promote a healthy ageing among older adults and improve employee well-being through a sustainable workplace culture [9,94,95,96], which is also in line with SDG eleventh (sustainable cities and communities). Last but not least, this study recommends future scholars to explore the suitable theoretical underpinnings not only in terms of human resource and social care practices, but also in terms of cultural, political, and economic aspects in ageing countries in order to expand the body of knowledge and gain a comprehensive understanding of how to retain eldercare workers in different contexts or specific regions.

## Figures and Tables

**Figure 1 ijerph-20-03932-f001:**
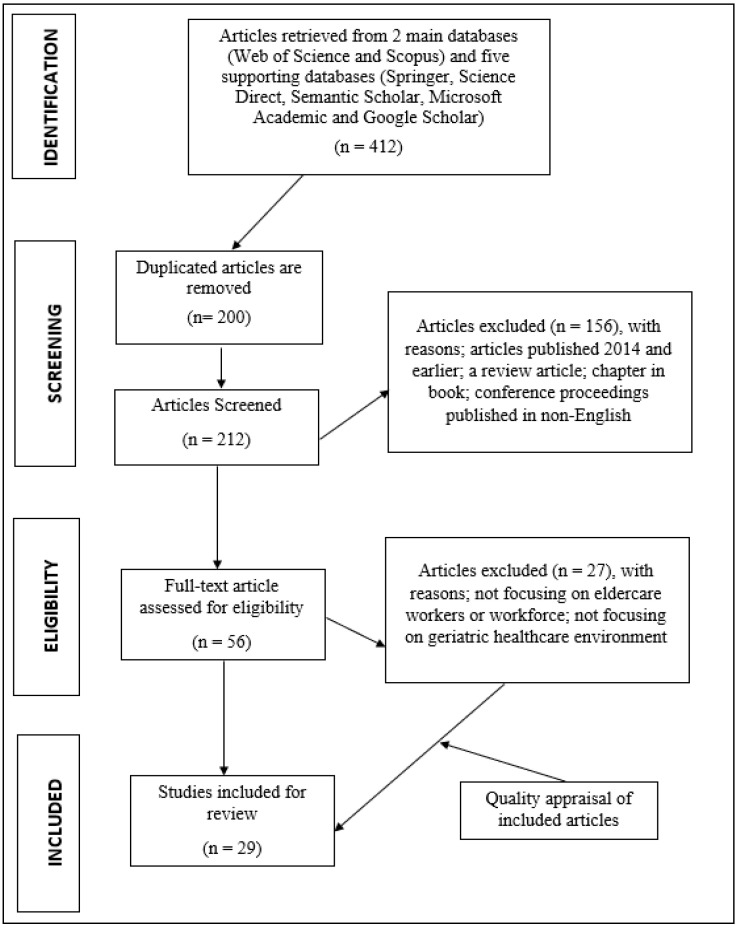
ROSES flow diagram of the searching process (Adapted from [21,22]).

**Figure 2 ijerph-20-03932-f002:**
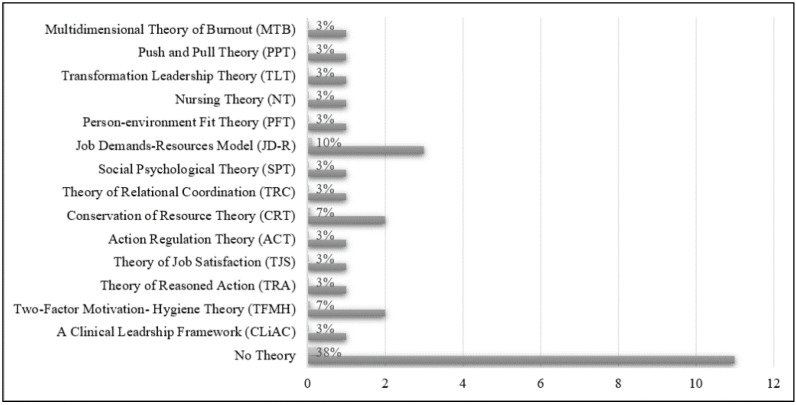
Underpinning theories/frameworks in the reviewed articles.

**Figure 3 ijerph-20-03932-f003:**
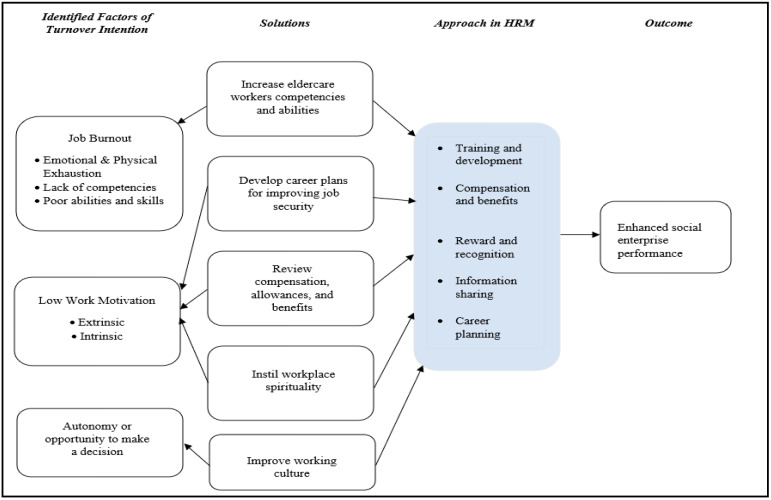
Study Framework.

**Table 1 ijerph-20-03932-t001:** Search strings for the main databases.

Database	Search String
Web of Science	TS = (intention to leave)) OR TS = (“turnover intentions”)) OR TS = (“obligation to stay”)) AND TS = (“employee retention”)) AND TS = (“nursing homes”)) OR TS = (“long term care”)) OR TS = (“old folks’ home”)) OR TS = (“eldercare”)) OR TS = (“eldercare social enterprise”) AND TS = (“geriatric workforce” OR “aged care worker” OR “eldercare worker” OR “registered nurse”)
Scopus	TITLE-ABS-KEY ((“turnover intentions” OR “intention to leave” OR “intention to stay” OR “retention”) AND (“long-term care” OR “nursing home” OR “elderly care” OR “Old folk home”) AND (“geriatric workforce” OR “aged care workers” OR “eldercare” OR “caregivers”))

**Table 2 ijerph-20-03932-t002:** Inclusion and exclusion criteria.

Criterion	Inclusion	Exclusion
Document Type	Articles with empirical data and full-text articles	Review articles, conference proceedings, chapter in books, book, non-full-text and duplicate publications
Timeline	2015–2021	2014 and earlier
Language	English	Non-English
Samples	Eldercare workers, geriatric workforce, or employees in long-term care	Non-employees or employees not working in geriatric settings or long-term care
Country coverage	Worldwide	N/A

Notes: N/A = Not Applicable.

**Table 3 ijerph-20-03932-t003:** Themes and sub-themes derived from the thematic analysis.

Theme		Factors of Turnover Intention						HR Approach					
		Job Burnout		Low Motivation		Job Autonomy		Abilities & Competencies		Job Motivation		Improve Working Cultures	Workplace Spirituality
Article	Sub-theme/year	PE	IC	EX	IN	MS	TS	MC	TD	EX	IN		
Jeon et al. [40]	2015		/			/	/		/			/	
Elliot et al. [35]	2015	/		/	/			/		/		/	
Austen et al. [31]	2016	/	/	/		/	/		/	/			
Rahnfeld et al. [27]	2016	/				/	/		/	/		/	
Kim & Kim [36]	2017	/		/		/		/					
Gao et al. [33]	2017					/	/					/	
Weale et al. [42]	2017		/	/					/				
Abd Aziz et al. [12]	2017	/		/		/			/	/		/	
Radford & Meissner [45]	2017	/	/	/		/	/		/	/		/	
Eltaybani et al. [29]	2018	/	/	/					/	/	/	/	
Ravenswood & Haar [28]	2018	/	/			/	/	/	/			/	
Cheng et al. [46]	2018	/	/	/					/	/	/	/	
Piers et al. [39]	2018						/	/				/	
Jakobsen et al. [34]	2018	/					/	/					
Strandell [18]	2019	/		/	/	/		/	/	/			/
Gaudenz et al. [32]	2019	/		/	/	/		/	/		/	/	
Wang et al. [41]	2019			/	/					/	/		
Xerri et al. [19]	2019	/		/	/	/							
Tei-tominaga & Nakanishi [21]	2020					/			/			/	/
Lee & Shin [47]	2020			/	/	/		/				/	
Chon & Kim [17]	2020	/	/	/					/	/			
Chen et al. [46]	2020	/		/					/	/			
Dhakal et al. [16]	2020	/	/	/					/	/		/	/
Xiao et al. [20]	2020	/				/		/	/			/	
Zhang et al. [13]	2020			/	/					/	/		
Xie et al. [38]	2020					/			/			/	
Lundmark et al. [37]	2020	/				/	/					/	
Lim [15]	2021	/	/	/	/	/	/		/				
Chang et al. [48]	2021	/							/				

**Table 4 ijerph-20-03932-t004:** Acronym of themes and sub-themes.

Job Burnout	Low Motivation	Job Autonomy	Abilities & Competencies	Job Motivation
PE = Physical and emotional burnout	EX = Extrinsic Motivation	MS- Managerial Support	MC—Manager competencies	EX-Extrinsic Motivation
IC = Incompetencies	IN = Intrinsic Motivation	TS = Teammates/Colleagues Support	TD = Training and development	IN = Intrinsic Motivation

## Data Availability

No new data was created or analysed in this study. Data sharing is not applicable to this article.

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
