# Peer review of "Eldercare’s Turnover Intention and Human Resource Approach: A Systematic Review"

_ijerph, 2023, doi:10.3390/ijerph20053932_

Round 1
Reviewer 1 Report
Certain references can be updated, and more current numbers would be helpful. Eg: United Nations Department of Economic and Social Affairs, Population 36 Division (2020).
A section on the significance of the study along with the data support would help to show the importance of the study.
Region of study (Sec 3.2): Studies related to ASEAN region may be helpful, as countries like Singapore have Ageing as a major problem and Eldercare requirements/ research are crucial. Examples:
A study on Singapore’s ageing population in the context of eldercare initiatives using machine learning algorithms - Big Data and Cognitive Computing
Smart eldercare in Singapore: Negotiating agency and apathy at the margins - Journal of Aging Studies
A study on impact of ageing population on Singapore healthcare systems using machine learning algorithms - World Review of Entrepreneurship, Management and Sustainable Development
Possible, that you would have been looking at the inclusion criteria of 2015–2021. May be latest 2022 studies would help especially Eldercare requirements may have changed due to COVID-19 situation.
Implications of the study to be provided with suggestions for the stakeholders involved.
Conclusions needs to be improved. Scope for future study can be false added.
Author Response
Dear Reviewer,
As attached rebutable article 2180860 for further action.
Thank you

Reviewer 2 Report
This article analyses a topical issue given that world population is getting older. With the evolution of medical care, it is natural to anticipate that this situation is going to have a dramatic increase, which is expected to produce a high demand for elder services. In addition, the employees that work at institution to support elder people developed a particularly great effort during the Covid 19 pandemic. The authors aim at to obtain the strategies used to retain elder workers and to determine what are the factors of turnover among them.
The structure of the article is chosen correctly and it exhibits a logical concern regarding the nature of the work developed that is a systematic literature review. The methodology is carefully applied. However, I have a few recommendations about the article:
I cannot find the reason or reasons to begin the analysis period on 2015. If the period was broader, the number of publications would be higher and it would give more quantity of information and allow to obtain more accurate results. In addition, the period analysed includes the Covid 19 pandemic that was not mentioned in the article. It would be important to analyse the impact of the pandemic in the turnover and retain of elderly care workers. For that purpose, I recommend the report of OECD 2020 “Who cares? Attracting and Retaining care workers for the elderly”.
In line 166 and based on previous text, it should be 27 and not 29 publications rejected. In paragraph beginning in line 227 the number of studier per country is 30 and not 29.
In my opinion, it would be important to distinguish factors that implies turnovers due to company origin or resulting from the nature of elder care work which may in turns just mean to change the company or by contrast leave the profession.
Author Response
Dear Reviewer,
As attached rebutable for reviewer 2nd for further action.
Thank you.

Reviewer 3 Report
I read your article with great interest. However, I am concerned about the following points, which led me to this opinion.
The World Health Organization's definition of older adults is 65 years and older, although studies from countries such as Germany, China, Sweden, Australia, Korea, Malaysia, and Japan were included in the survey. However, these countries do not necessarily define 65 years of age as older adults. I need to clarify this point before discussing the results.
In order to clarify the research questions of this paper, it is necessary to consider the economic situation, social security system, religion, population, and average life expectancy of each country.
You mainly searched "organizational behavior and human resource management, geriatrics and gerontology, social sciences, and care planning," but have you investigated findings in areas such as economics, business administration, cultural anthropology, social welfare, and psychology? This point was unclear. Also, the fact that "PubMed" was not used for the search was also questionable.
Without clarification of these points, I believe that the bias in the results obtained in this review cannot be denied.
Author Response
Dear Editor,
As attached file for all reviewers comment for further action.
Thank you

Round 2
Reviewer 2 Report
This article is focused on a very important topic issue. Therefore it is an important contribution to the research for improving elder care services.
Author Response
Dear Reviewer,
As attached article improvement matrix for further action.
TQ

Reviewer 3 Report
The important indicator for this study is the countries who are approaching or currently experiencing challenges on becoming an ageing nation. Moreover, the scope of paper is eldercare worker, not older adults.
Regarding the above comment.
Are the jobs of eldercare workers the same in countries with long life expectancies and those without? For example, in countries with a longer life expectancy, they spend more time needing care, so knowledge about " advance care planning " and " palliative care " is required. Do these not affect the turnover intention of the subject? Even if the eldercare worker is not an elderly person, these points need to be clarified.
Author Response

(The authors gave the same response as above.)
